# Involvement of AMPKα and MAPK-ERK/-JNK Signals in Docetaxel-Induced Human Tongue Squamous Cell Carcinoma Cell Apoptosis

**DOI:** 10.3390/ijms232213857

**Published:** 2022-11-10

**Authors:** Chin-Chuan Su, Jhe-Wei Lin, Kai-Yao Chang, Cheng-Tien Wu, Shing-Hwa Liu, Kai-Chih Chang, Jui-Ming Liu, Kuan-I Lee, Kai-Min Fang, Ya-Wen Chen

**Affiliations:** 1Department of Otorhinolaryngology, Head and Neck Surgery, Changhua Christian Hospital, Changhua County, Changhua County 500, Taiwan; 2Department of Physiology, School of Medicine, College of Medicine, China Medical University, Taichung 404, Taiwan; 3Department of Emergency, Taichung Tzu Chi Hospital, Buddhist Tzu Chi Medical Foundation, Taichung 427, Taiwan; 4Department of Nutrition, China Medical University, Taichung 404, Taiwan; 5Institute of Toxicology, College of Medicine, National Taiwan University, Taipei 100, Taiwan; 6Center for Digestive Medicine, Department of Internal Medicine, China Medical University Hospital, Taichung 404, Taiwan; 7Department of Urology, Taoyuan General Hospital, Ministry of Health and Welfare, Taoyuan 330, Taiwan; 8Department of Obstetrics and Gynecology, Tri-Service General Hospital, National Defense Medical Center, Taipei 114, Taiwan; 9Department of Otolaryngology, Far Eastern Memorial Hospital, New Taipei City 220, Taiwan

**Keywords:** docetaxel, tongue squamous cells carcinomas, apoptosis, AMPKα, ERK, JNK

## Abstract

Cancers of the oral cavity can develop in the anatomic area extending from the lip, gum, tongue, mouth, and to the palate. Histologically, about 85–90% of oral cavity cancers are of the type squamous cells carcinomas (SCCs). The incidence of oral tongue SCC is higher in the tongue than any other anatomic area of the oral cavity. Here, we investigated the therapeutic effects and molecular mechanisms of docetaxel, which is a paclitaxel antitumor agent, on the cell growth of a human tongue SCC-derived SAS cell line. The results showed that docetaxel (10–300 nM) induced cytotoxicity and caspase-3 activity in SAS cells. Moreover, docetaxel (100 nM) promoted the expression of apoptosis-related signaling molecules, including the cleavages of caspase-3, caspase-7, and poly (ADP-ribose) polymerase (PARP). In mitochondria, docetaxel (100 nM) decreased the mitochondrial membrane potential (MMP) and Bcl-2 mRNA and protein expression and increased cytosolic cytochrome c protein expression and Bax mRNA and protein expression. In terms of mitogen-activated protein kinase (MAPK) and adenosine monophosphate-activated protein kinase (AMPK) signaling, docetaxel increased the expression of phosphorylated (p)-extracellular signal-regulated kinase (ERK), p-c-Jun N-terminal kinase (JNK), and p-AMPKα protein expression but not p-p38 protein expression. Moreover, the increase in caspase-3/-7 activity and Bax protein expression and decreased Bcl-2 protein expression and MMP depolarization observed in docetaxel-treated SAS cells could be reversed by treatment with either SP600125 (a JNK inhibitor), PD98059 (an MEK1/2 (mitogen-activated protein kinase kinase 1/2) inhibitor), or compound c (an AMPK inhibitor). The docetaxel-induced increases in p-JNK, p-ERK, and p-AMPKα protein expression could also be reversed by treatment with either SP600125, PD98059, or compound c. These results indicate that docetaxel induces human tongue SCC cell apoptosis via interdependent MAPK-JNK, MAPK-ERK1/2, and AMPKα signaling pathways. Our results show that docetaxel could possibly exert a potent pharmacological effect on human oral tongue SCC cell growth.

## 1. Introduction

Cancers of the oral cavity can develop in the anatomic area extending from the lip, gum, tongue, mouth, and to the palate [1]. Around the world, the annual incidence of such cancers has been increasing, with over 377,713 diagnosed cases in 2020 and an annual mortality of about 177,757 deaths [2]. Histologically, squamous cell carcinomas (SCCs) are the most common type of oral cavity cancer, representing 85–90% of cases [1,3,4,5]. Moreover, in a study of Scandinavian tumor registries, the incidence of oral tongue SCC was 0.06–0.32 per 100,000 (about 5-fold) among young men and 0.03–0.19 per 100,000 (about 6-fold) among young women compared to only a 2-fold in older age groups [6]. The risk factors of oral SCC are associated with smoking, alcohol consumption, human papilloma virus (HPV) infection, and ultraviolet radiation [7,8,9,10]. The lesions damage the functions of speech, swallowing, and chewing, and induce pain, which will seriously affect patients in their daily life [10,11]. Current methods for treating oral tongue SCC include surgery and radiotherapy. However, the survival rate remains poor, and with a high incidence of metastasis [10,11,12,13]. Thus, the development of new therapeutic strategies for tongue SCC is urgent and necessary.

Docetaxel is a semisynthetic, second-generation paclitaxel antitumor agent, which was derived from European yew tree. The pharmacological effect of docetaxel is to bind and stabilize tubulin, thereby interfering with the microtubule network during mitosis. This results in cell cycle arrest at the G2/M phase and subsequent induction of cell death [14,15,16]. Docetaxel has been investigated in therapy of many cancer diseases, for example, salivary gland carcinoma, non-small cell lung cancer, and prostate cancer [16,17,18].

Previously, research in murine salivary carcinoma cells has demonstrated that docetaxel induces apoptosis by altering Bcl-2 family and MAPK signals. The c-Jun N-terminal kinase/stress-activated protein kinase (JNK/SAPK) is necessary for docetaxel-induced cell cycle arrest [19]. Induction of JNK activation has been shown to up-regulate Bak and Bax and promote apoptosis [19,20,21,22]. Furthermore, AMPK is a serine/threonine kinase that consists of three subunits including catalytic α, scaffolding β, and AMP-sensing γ in mammalian cells [23,24]. It has been reported that the decrease of cellular ATP would cause the AMP/ATP ratio to increase. This effect results in the AMP binding to AMPK, which caused the upstream kinases, such as LKB1 and CAMKKβ, to phosphorylate the AMPKα subunits on the Thr172 [25,26,27,28]. Many studies have suggested that AMPK activation regulates cancer cell proliferation through reprogramming of cell metabolism [24,29,30]. One study demonstrated that the reduction in AMPK activation of breast cancer specimens is associated with higher histological grade and axillary node metastasis [31]. Another study also showed that a decrease in AMPKα expression can be observed in the non-small cell lung cancer, and another showed that AMPKα activation can inhibit lung cancer cell growth [26]. Together, these studies suggest that AMPKα serves an important role in cancer prevention and treatment. Despite growing evidence, little is known about the role of AMPKα in oral tongue SCC. The aim of the present study is to evaluate the roles of AMPKα and MAPKs signals in human oral tongue SCC cell growth and to determine whether docetaxel can reduce oral tongue SCC cell growth through the regulation of AMPKα and MAPKs signals.

## 2. Results

### 2.1. Docetaxel Induces Cytotoxicity and Apoptosis in Human Tongue SCC SAS Cells

We first investigated whether docetaxel induces cytotoxicity and apoptosis in SAS cells. Cells were treated with docetaxel (10–300 nM) for 24 and 48 h. As shown in Figure 1A, docetaxel (100 and 300 nM; for 48 h) markedly induced morphological changes, such as cell shrinkage, as observed by inverted phase-contrast microscopy. Furthermore, cell viability was significantly decreased in a concentration- and time-dependent manner following treatment with docetaxel (10–300 nM) (Figure 1B). The 50% lethal concentration (LC50) in SAS cells was ~100 nM docetaxel treatment at 48 h (Figure 1B). In order to clarify whether docetaxel induces apoptosis in SAS cells, caspase-3 activity was detected. Cells were pretreated with casepase-3 inhibitor (Z-DEVD-FMK) for 1 h and then with docetaxel (10 and 100 nM) for 24 and 48 h. The caspase-3 activity was significantly increased by docetaxel treatment, which could be significantly inhibited by Z-DEVD-FMK pretreatment (Figure 1C). Moreover, the caspase-related signals, including cleaved casepase-3, cleaved caspase-7, and cleaved PARP, were increased after docetaxel 100 nM treatment for 24 and 48 h (Figure 1D). These results indicate that docetaxel can induce apoptosis in SAS cells.

### 2.2. Docetaxel-Induced Apoptosis Is Mediated by a Mitochondria-Dependent Pathway in Human Tongue SCC SAS Cells

In the next experiments, we determined the possible mechanisms of docetaxel-induced apoptosis in SAS cells. Cells were treated with docetaxel (100 nM) for 24, 36, and 48 h. The results showed that the MMP was significantly decreased after docetaxel treatment (Figure 2A). Cytosolic cytochrome c protein expression was significantly increased at 48 h of docetaxel administration (Figure 2B). Furthermore, docetaxel significantly increased pro-apoptotic *Bax*, *Bak*, *Bad*, and *p53* (Figure 2C) and decreased anti-apoptotic *Bcl-2* (Figure 2D) gene expression levels. As shown in Figure 2E–G, treatment with docetaxel for 48 h also significantly decreased Bcl-2 and increased Bax protein expression in SAS cells. These showed a significant shift in the pro-apoptotic/anti-apoptotic ratio toward a state associated with apoptosis. These results indicate that mitochondria-dependent apoptotic pathway plays an important role in docetaxel-induced SAS cell death.

### 2.3. Docetaxel Induces Phosphorylation of MAPK-ERK/JNK and AMPKα in Human Tongue SCC SAS Cells

We next investigated whether MAPK and AMPKα signals were involved in docetaxel-induced SAS cell apoptosis. After cells were treated with docetaxel (100 nM) for 1 to 24 h, the protein expression of p-ERK1/2, p-JNK1/2, and p-AMPKα, but not that of p-p38, was significantly increased (Figure 3). In order to determine the roles of ERK, JNK, and AMPKα in docetaxel-induced cell apoptosis, cells were pretreated with SP600125 (a JNK inhibitor; 20 μM), PD98059 (an MEK1/2 inhibitor; 20 μM), or compound c (an AMPK inhibitor; 10 μM) for 1 h, and then treated with or without docetaxel for 48 h. The results showed that docetaxel-induced caspase-3 activity could be significantly inhibited by pretreatment with SP600125, PD98059, or compound c (Figure 4A). These results indicate that ERK, JNK, and AMPKα signals are involved in the docetaxel-induced SAS cell apoptosis. Moreover, SP600125, PD98059, and compound c pretreatment significantly reversed the docetaxel-induced MMP loss (Figure 4B), decreased Bcl-2 protein expression, and increased Bax protein expression (Figure 2C–E). Pretreatment with SP600125, PD98059, or compound c could significantly inhibit the docetaxel-induced protein expression of p-JNK, p-ERK, and p-AMPKα (Figure 5A–C). Thus, these results suggest that the activations of JNK, ERK1/2, and AMPKα signals induced by docetaxel are interdependent, which regulates the mitochondria-dependent apoptotic pathway downstream, resulting in tongue SCC SAS cell death.

## 3. Discussion

Squamous cell carcinomas (SCC) are the major histological types of oral cancer, and their development is associated with malignancy and a lower survival rate in patients [32,33]. It was recently shown that the incidence of oral cancer has been decreasing in anatomic areas of the lip, gum, mouth, and palate, but not in the tongue, leading to the suggestion that there are possible etiological or genetic factors triggering carcinogenesis in the tongue that are linked with certain geographic areas and differences in sex [32,33,34,35]. There are many risk factors involved in tongue SCC disease formation, for example, tobacco, betel nut chewing, and alcohol. The incidence rate of tongue SCC is higher in older men with exposure to the aforementioned risk factors. A report showed the incidence of tongue SCC increasing at the ages of 18 to 44 in young white women [36]. Moreover, the incidence of tongue SCC varies by gender in different countries; it is higher in men in Austria, Bulgaria, and Ireland but higher in women in England, Denmark, Sweden, and the United States [32]. Thus, many etiological factors might drive tongue SCC programming. More importantly, the development of new therapeutic strategies for tongue SCC is urgent and necessary.

Docetaxel is a semisynthetic derivative of 10-deactyl baccatin III from the European yew. It is currently approved in treatment of patients with SCC of the head and neck [37,38]. However, in the current literature, there is a lack of solid data to suggest the possible molecular mechanisms of docetaxel on tongue SCCs.

Inducing tumor cell death is a therapeutic strategy used for many cancer diseases. The apoptotic effect has been shown to induce cell shrinkage from generation of pyknotic nucleus and chromatin condensation, leading to induction of apoptosis with cell nuclear collapse and apoptotic body formation [39]. Apoptosis involves extrinsic (receptor-dependent) and intrinsic (mitochondrial-dependent) pathways. Whether by the extrinsic or intrinsic pathway, apoptotic signals induce mitochondrial transmembrane potential disruption and trigger apoptotic factors such as DNA condensation, DNA degradation, and apoptosome formation [40,41,42]. It has been reported that the Bcl-2 family regulates mitochondrial permeabilization and leads to the release of apoptogenic factors that activate caspases and apoptosis [43]. In the Bcl-2 family, the BH3 domain of activator BH3-only proteins binds to the BH3 domain-binding groove in Bax/Bak. Bax/Bak activation results in conformational changes to the mitochondrial outer membrane and permeability transition pore formation [43]. Moreover, cytochrome c is synthesized in the cytoplasm and translocated through the outer mitochondrial membrane, which leads to assembly of the apoptosome. The apoptosome comprises cytochrome c, Apaf-1, and dATP. Cytochrome c binds to dATP, resulting in conversion of Apaf-1 from a closed monomeric configuration to an open heptameric platform for assembly and activation of caspases [41,44]. In our results, docetaxel induced cytotoxic cell death and caspase-3 activity in SAS cells. Regarding apoptotic signals, docetaxel increased the cleavages of the caspase-3, caspase-7, and PARP proteins. These results indicate that docetaxel induces the apoptosis of SAS cells. Studies have shown that docetaxel induces apoptosis mainly through the mitochondria-dependent pathway [40,41,42]. Our results found that docetaxel decreased mitochondrial transmembrane potential, increased cytosolic cytochrome c protein expression, increased *Bax*, *Bak*, *Bad*, and *p53* mRNA expression, and decreased *Bcl-2* mRNA expression in SAS cells.

MAPK signals are reported to be associated with cell proliferation, cell differentiation, apoptosis, angiogenesis, invasion, and metastasis [45]. MAPKs include ERK1/2, JNK, p38, and ERK5 [45,46]. Activating ERK1/2 regulates pro-apoptotic signaling and induces apoptosis. One study showed that cisplatin induces DNA damage-induced neuronal cell apoptosis by increasing the protein expression of p-ERK1/2 and p53 [47]. Another study also showed that perfluorohexanesulfonate (PFHxS) induces apoptosis of neuronal cells and phosphorylation of ERK1/2, JNK, and p38 proteins [48]. It was further found that ERK1/2 inhibition reduces apoptosis, while JNK inhibition increases apoptosis; however, p38 inhibition does not affect apoptosis [48]. In oral squamous carcinoma cells, JNK activation can regulate apoptosis through the mitochondria and ER stress signals [49]. These findings indicate that MAPK signaling is associated with the apoptotic pathway. In this study, we found that docetaxel increased the expression of p-ERK1/2 and p-JNK but not p-p38 protein. Both SP600125 (a JNK inhibitor) and PD98059 (an MEK1/2 inhibitor) reversed docetaxel-induced caspase-3/-7 activity and MMP depolarization, increased Bax expression, and decreased Bcl-2 expression in SAS cells.

The activation of AMPK has been reported to inhibit non-small cell lung cancer cell growth [26]. Previous studies have shown that higher concentrations (over 10–100 μM) of traditional AMPK activators, such as AICAR and metformin, can induce anticancer effects in non-small cell lung carcinoma cells [50,51,52,53]. In an oral cancer study, the upregulation of p-AMPKα (Thr172), p-AMPK𝛼1/𝛼2 (Ser425/Ser491), p-AMPKβ1 (Ser108), and p-AMPK𝛾1 could modulate the invasion and migration of oral cancer cells [54]. In our study, we found that docetaxel could induce tongue SCC cell death at the concentrations of 10–300 nM and induce AMPKα activation at a 100 nM concentration. Pretreatment with an AMPK inhibitor (compound c) effectively prevented AMPKα activation and inhibited docetaxel-induced caspase-3/-7 activity and MMP depolarization, increased Bax expression, and decreased Bcl-2 expression in SAS cells as well as the activation of JNK and ERK1/2 signals. Moreover, the phosphorylation of JNK, as well as ERK1/2 and AMPKα, in docetaxel-treated SAS cells was abrogated by pretreatment with SP600125 (a JNK inhibitor); meanwhile, the phosphorylation of ERK1/2, as well as JNK and AMPKα, was also markedly inhibited by pretreatment with PD98059 (an MEK1/2 inhibitor). These results imply that JNK, ERK1/2, and AMPKα-mediated signals are interdependent and play critical roles in the downstream regulation of the mitochondria-dependent apoptosis pathway in docetaxel-induced tongue SCC cell death.

Limitations of the study: (1) A human tongue cancer SAS cell line was used in this study. Other tongue cancer cell lines can be used to further confirm the docetaxel-mediated molecular mechanisms. (2) The pharmacological inhibitors for JNK, ERK, and AMPK were used in this study. JNK-, ERK-, and AMPK-specific targeted gene silencing can be used to further confirm the molecular mechanisms mediated by docetaxel. (3) Future animal model studies can further support these findings from an in vitro cancer cell model.

## 4. Materials and Methods

### 4.1. Materials

Docetaxel and other chemicals (including SP600125, PD98059, and compound c), unless specified otherwise, were purchased from Sigma-Aldrich (St. Louis, MO, USA). Docetaxel was dissolved in dimethyl sulfoxide (DMSO) to prepare the stock solution. The maximum volume of DMSO used in the experiments was <1% per well, which does not induce any cytotoxicity. Laboratory plastic ware was obtained from Falcon (Becton, Dickinson and Company, Franklin Lakes, NJ, USA). Dulbecco’s modified Eagle’s medium and Ham’s F12 medium, fetal bovine serum (FBS), and antibiotics (penicillin-streptomycin (Cat. No.: 15140122) and gentamicin (Cat. No.: 15710064)) were purchased from Gibco/Invitrogen (Thermo Fisher Scientific Inc., Waltham, MA, USA). Mouse or rabbit monoclonal antibodies specific for cleaved caspase-3 (Cat. No.: #9661), cleaved caspase-7 (Cat. No.: #9491), PARP (Cat. No.: #9542), cytochrome c (Cat. No.: #11940), Bcl-2 (Cat. No.: #15071), Bax (Cat. No.: #89477), phosphorylated (p)-JNK (Cat. No.: #9255), p-ERK1/2 (Cat. No.: #4377), p-p38 (Cat. No.: #9216), p-AMPKα (Cat. No.: #4188), JNK-1 (Cat. No.: #3708), ERK1/2 (Cat. No.: #9102), p38 (Cat. No.: #8690), AMPKα (Cat. No.: #2532), β-actin (Cat. No.: #8457), and secondary antibodies (horseradish peroxidase (HRP), conjugated anti-mouse IgG (Cat. No.: #7076), or anti-rabbit IgG (Cat. No.: #7074)) were purchased from Cell Signaling Technology (Danvers, MA, USA).

### 4.2. Cell Culture

The human tongue SCC-derived SAS cell line (JCRB0260) was purchased from the Japanese Collection of Research Bioresources Cell Bank. SAS cells were cultured in a humidified chamber containing a 5% CO_2_—95% air mixture at 37 °C. Cells were maintained in culture medium containing 45% Dulbecco’s modified Eagle’s medium and 45% Ham’s F12 medium supplemented with 10% FBS, 1% penicillin-streptomycin, and 0.5% gentamicin. Cells were seeded into 6-, 24-, or 96-well culture plates for each experiment and allowed to grow for 12–18 h (with recovery overnight) and then treated with or without docetaxel (10–300 nM) for various time intervals in the absence or presence of the inhibitors of SP600125 (20 μM), PD98059 (20 μM), or compound c (10 μM) for 1 h at 37 °C prior to treatment with docetaxel.

### 4.3. Morphological Analysis

The changes in cell morphology were detected according to a previous study [55]. SAS cells were cultured on a glass slide at a density of 1 × 10^6^ cells/well at 37 °C. After 24 h, a photomicrograph was obtained with a 20× objective lens using a cooled CCD camera attached to a Zeiss Axiovert 135-TV Inverted Fluorescence Phase Microscope (Carl Zeiss AG, Germany).

### 4.4. Cytotoxicity Assay

SAS cells were washed with fresh medium and cultured in 96-well plates (2 × 10^4^ cells/well) and then treated with or without docetaxel (10–300 nM) for 24 h. After incubation, the medium was aspirated and cells were incubated with fresh medium containing 0.2 mg/mL 3-(4,5-dimethyl thiazol-2-yl-)-2,5-diphenyl tetrazolium bromide (MTT). After 4 h, the medium was removed, and blue formazan crystals were dissolved in 100 μL of DMSO. The absorbance at 570 nm was measured using a Bio-Tek uQuant™ Microplate Reader (MTX Lab Systems, Winooski, VT, USA).

### 4.5. Determination of Caspase-3 Activity

Caspase-3 activity was assessed using a Caspase-3 Activity Assay Kit (Cell Signaling Technology, Inc., Danvers, MA, USA). SAS cells were seeded at 2 × 10^5^ cells/well in a 24-well plate and treated with or without docetaxel (10 and 100 nM) in the absence or presence of Z-DEVD-FMK (10 μM) at 37 °C. At the end of treatment (24 and 48 h), the cell lysates were incubated at 37 °C with 10 μM Ac-DEVD-AMC, a caspase-3/CPP32 substrate, for 1 h. The fluorescence of the cleaved substrate was measured using a spectrofluorometer (Gemini XPS Microplate Reader, Molecular Devices, San Jose, CA, USA) at an excitation wavelength of 380 nm and an emission wavelength of 460 nm.

### 4.6. Detection of Mitochondrial Membrane Potential (MMP)

MMP was analyzed using a fluorescent probe 3,3′-dihexyloxacarbocyanine iodide (DiOC_6_), which was a positively charged mitochondria-specific fluorophore. Briefly, SAS cells were seeded at 2 × 10^5^ cells/well in a 24-well plate and incubated with or without docetaxel (100 nM) for 24–48 h. At the end of treatment, cells were incubated with medium containing 100 nM DiOC_6_ for 30 min at 37V °C. After incubation with the dye, cells were harvested and washed twice with phosphate-buffered saline (PBS) and then re-suspended in ice-cold PBS. MMP was analyzed by a flow cytometer (FACScalibur, Becton, Dickinson and Company, USA) using CellQuest software version 5.1 (Becton, Dickinson and Company).

### 4.7. Caspase -3/-7 Activity Assay

Caspase-3/-7 activity was determined using a FLICA Caspase-3/-7 assay kit (ImmunoChemistry Technologies, LLC, CA, USA). SAS cells were seeded at 5 × 10^5^ cells/well in a 24-well plate and incubated with or without docetaxel (100 nM) for 48 h in the absence or presence of SP600125 (20 μM), PD98059 (20 μM), or compound c (10 μM) at 37 °C (1 h pretreatment). At the end of treatment, cells were washed twice with PBS and stained with fluorescent probes for 30 min in a dark environment at 37 °C. After incubation with the dye, cells were carefully removed supernatants and washed twice with 1× wash buffer. The fluorescence intensity was measured using a spectrofluorometer (Gemini XPS Microplate Reader, Molecular Devices, San Jose, CA, USA) at an excitation wavelength of 488 nm and an emission wavelength of 530 nm.

### 4.8. Western Blot Analysis

SAS cells were seeded at 1 × 10^6^ cells/well in a 6-well culture plate treated with or without docetaxel in the absence or presence of SP600125 (20 μM), PD98059 (20 μM), or compound c (10 μM) at 37 °C (1 h pretreatment). At the end of treatment for various durations, protein expression levels were analyzed by Western blotting as previously described [55,56]. In brief, equal amounts of protein (50 μg per lane) were subjected to electrophoresis on 10% (*w*/*v*) SDS-polyacrylamide gels and transferred to polyvinylidene difluoride (PVDF) membranes. The membranes were blocked for 1 h in PBST (PBS, 0.05% Tween-20) containing 5% nonfat dry milk. After blocking, the membranes were incubated with mouse or rabbit monoclonal antibodies specific for cleaved caspase-3, cleaved caspase-7, cleaved PARP, p-JNK1/2, JNK-1, p-ERK1/2, ERK, p-p38, p38, p-AMPKα, AMPKα, and β-actin in 0.1% PBST (1:1000) for 12–16 h at 4 °C. After 3 additional washes in 0.1% PBST (15 min each), the respective HRP-conjugated secondary antibodies were applied (1:2500 in 0.1% PBST) for 1 h at 4 °C. The antibody-reactive bands were developed using enhanced chemiluminescence reagents (PierceTM, Thermo Fisher Scientific Inc.) and analyzed by a luminescent image analyzer (ImageQuant™ LAS-4000, GE Healthcare Bio-Sciences Corp., Piscataway, NJ, USA). For cytosol cytochrome c expression, the detection was performed as previously described by Chang et al. [34]. In brief, at the end of treatments cells were detached, washed twice with PBS, and then homogenized using a mortar and pestle in the extract buffer (0.4 M mannitol, 25 mM 3-(N-morpholino)propanesulfonic acid (MOPS; pH 7.8), 1 mM ethylene glycol bis(2-aminoethyl ether)-N,N,N’,N’-tetraacetic acid (EGTA), 8 mM cysteine, and 0.1 % (*w*/*v*) bovine serum albumin). The cell debris was removed via centrifugation at 6000× *g* for 2 min. The supernatant was centrifuged again at 12,000× *g* for 15 min. The supernatant (cytosolic fraction) was used to detect cytochrome c expression by Western blot analysis.

### 4.9. Real-Time Quantitative Reverse-Transcription Polymerase Chain Reaction (RT-qRT-PCR) Analysis

The expression of apoptosis-related genes was evaluated by RT-qRT-PCR analysis as previously described [55,56]. Briefly, intracellular total RNA was extracted using RNeasy kits (Qiagen, Hilden, Germany) and reverse-transcribed into cDNA using AMV RTase, a reverse transcriptase enzyme (Promega Corporation, Madison, WI, USA), according to the manufacturer’s instructions. Each sample (2 μL cDNA) was then assayed with Real-Time SYBR Green PCR reagent (Invitrogen, USA) in a 25 μL reaction volume and the following specific human primers: Bcl-2 forward (5′-TTAGATCTATGGCGCACGCTGGGAGAAC-3′) and reverse (5′-CGAATTCTCACTTGTGGCTCAGATAGG-3′) [57]; Bax forward (5′-CTTTTGCTTCAGGGTTTCATCC-3′) and reverse (5′-TTGAGACACTCGCTCAGCTTCT -3′) [58]; Bak forward (5′-GCCCAGGACACAGAGGAGGTTTTC-3′) and reverse (5′-AAACTGGCCCAACAGAACCACACC-3′) [59]; Bad forward (5′-CGGAGGATGAGTGACGAGTT-3′) and reverse (5′-GATGTGGAGCGAAGGTCACT-3′) [60]; p53 forward (5′-CCTCACCATCATCACACTGG-3′) and reverse (5′-CCTCATTCAGCTCTCGGAAC-3′) [61]; β-actin forward (5′-GGCGACGAGGCCCAGA-3′) and reverse (5′-CGATTTCCCGCTCGGC-3′) [55]. The amplification was performed using an ABI StepOnePlus™ Sequence Detection System (Applied Biosystems, Thermo Fisher Scientific, Inc.). The cycling conditions were 10 min of polymerase activation at 95 °C followed by 40 cycles at 95 °C for 15 s and 60 °C for 60 s. Real-time fluorescence detection was performed during the 60 °C annealing/extension step of each cycle. Melt curve analysis was performed on each primer set to ensure that no primer dimers or nonspecific amplifications were present under the optimized cycling conditions. After 40 cycles, data analysis was performed using StepOneTM software version 2.1 (Applied Biosystems, Thermo Fisher Scientific, Inc.). All amplification curves were analyzed with a normalized reporter (R_n_, ratio of fluorescence emission intensity to fluorescence signal of passive reference dye) and a threshold of 0.2 to obtain the C_T_ values (threshold cycle). The reference control genes were measured with four replicates in each PCR run, and their average CT was used for relative quantification analysis [62]. The expression data were normalized by subtracting the mean value of reference gene C_T_ from the C_T_ value (ΔC_T_). The fold change value was calculated using the 2^−ΔΔCT^ method, where ΔΔCT represents ΔC_T-condition of interest_ – ΔC_T-control_. Prior to conducting statistical analysis, the fold change from the mean of the control group was calculated for each individual sample.

### 4.10. Statistical Analysis

Data are presented as the mean ± standard deviation (SD) of at least three independent experiments. All data analyses were performed using the SPSS software version 12.0 (SPSS, Inc., Chicago, IL, USA). For each experimental condition, significant differences were assessed by one-way analysis of variance (ANOVA) followed by Tukey’s post hoc test; *p* value < 0.05 was considered to indicate a significant difference.

## 5. Conclusions

The results of this study, as shown in Figure 6, demonstrate that interdependent MAPK-JNK, MAPK-ERK1/2, and AMPKα signaling pathways are involved in docetaxel-induced mitochondria-dependent human tongue SCC cell apoptosis and death. These in vitro findings suggest that docetaxel possesses pharmacological potential for oral tongue SCC cancer therapy. The findings of cell model studies may be further explored in future animal model studies to elucidate the mechanism of action of docetaxel.

## Figures and Tables

**Figure 1 ijms-23-13857-f001:**
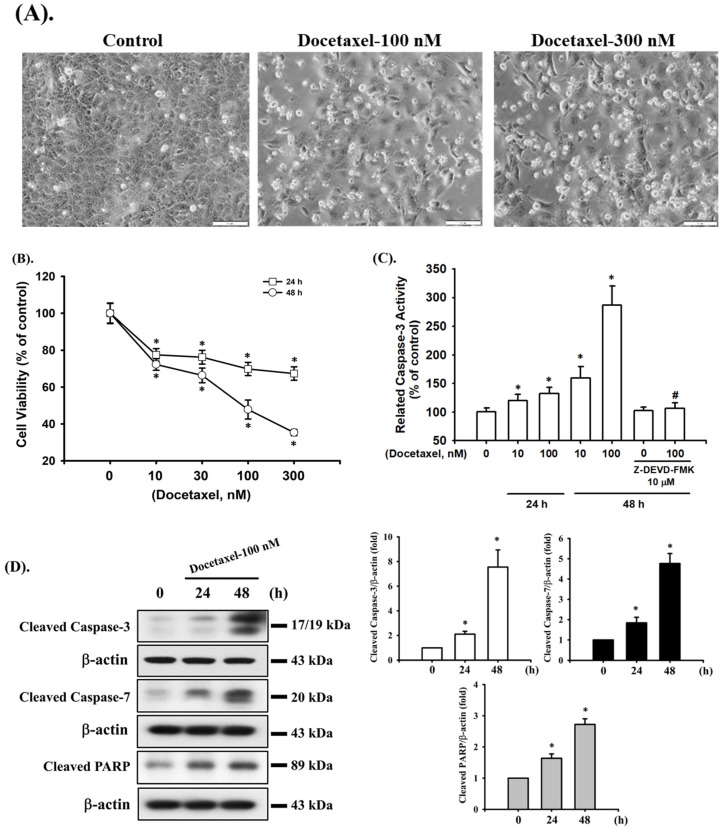
Effects of docetaxel on cell viability and apoptosis in human tongue squamous carcinoma SAS cells. (**A**) Cells were treated with docetaxel (100 and 300 nM) for 48 h. The morphological changes were observed using inverted phase-contrast microscopy (magnification ×200; scale bar = 100 μm). (**B**) Cells were treated with docetaxel (10–300 nM) for 24 and 48 h. The cell viability was determined by MTT assay. (**C**) Cells were pretreated with or without Z-DEVD-FMK (10 μM) for 1 h, and then treated with docetaxel (10 and 100 nM) for 24 or 48 h. The caspase-3 activity was determined using a caspase-3 activity assay kit. (**D**) Cells were treated with docetaxel (100 nM) for 24 and 48 h. The cleaved caspase-3, caspase-7, and PARP protein expression levels were determined by Western blot assay. Data in (**B**,**C**) are presented as the means ± SD of four independent experiments with triplicate determination. Results are shown in (**D**) with representative images, and quantification was performed by densitometric analysis. Each presented bar is the mean ± SD of three independent experiments. Statistical analysis was performed with one-way ANOVA analysis followed by Tukey’s post hoc test. * *p* < 0.05 compared with vehicle control. # *p* < 0.05 compared with docetaxel treatment group.

**Figure 2 ijms-23-13857-f002:**
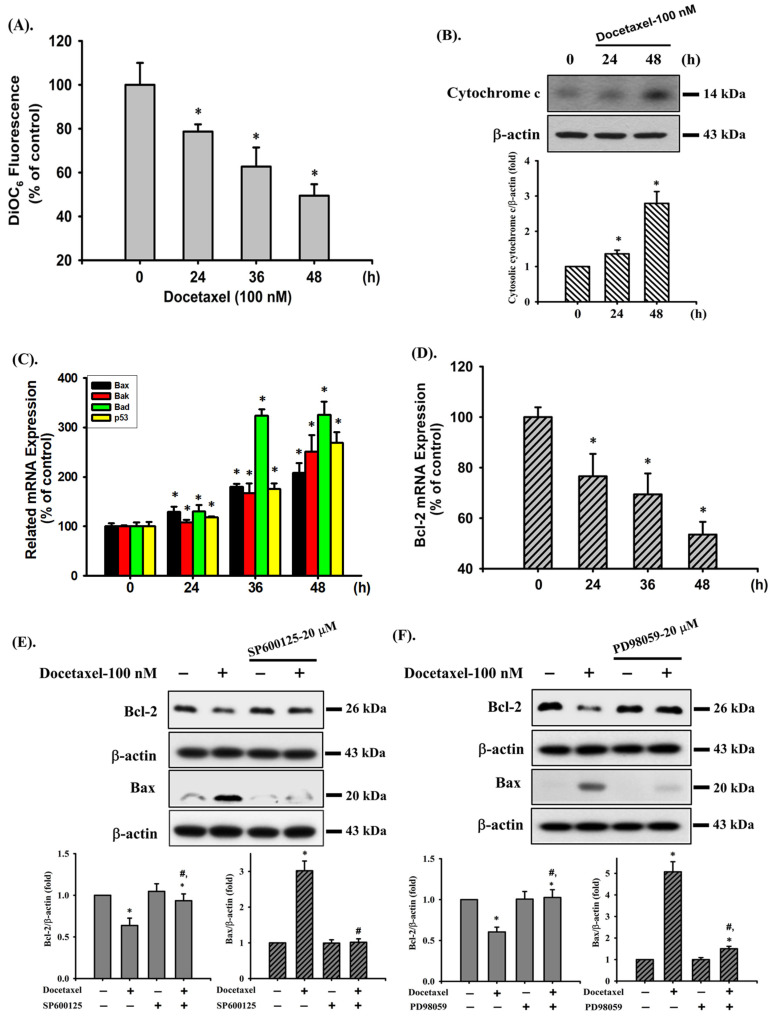
Effects of docetaxel on mitochondrial transmembrane potential (MMP), cytosolic cytochrome c protein expression, and mitochondrial damage-related mRNA expression in human tongue squamous carcinoma-derived SAS cells. (**A**) Cells were treated with docetaxel (100 nM) for 24, 36, and 48 h. MMP was determined by flow cytometer with a fluorescent dye DiOC_6_-FITC. (**B**) Cells were treated with docetaxel (100 nM) for 24 and 48 h. Cytosolic cytochrome c protein expression was determined by Western blot analysis. Cells were treated with docetaxel (100 nM) for 24, 36, and 48 h. The mRNA expression of (**C**) *Bax*, *Bak*, *Bad*, and *p53* and (**D**) *Bcl-2* was determined by RT-qRT-PCR analysis. Additionally, SAS cells were pretreated with SP600125 (a JNK inhibitor; 20 μM), PD98059 (an MEK1/2 inhibitor; 20 μM), and compound c (an AMPK inhibitor; 10 μM) for 1 h and then treated with or without docetaxel for 48 h, and (**E**,**F**,**G**) the protein expression of Bcl-2 and Bax was determined by Western blot analysis. Data in (**A**,**C**,**D**) are presented as the means ± SD of four independent experiments with triplicate determination. Results are shown in (**B**,**E**–**G**) with representative images, and quantification was performed by densitometric analysis. Each presented bar is the mean ± SD of three independent experiments. Statistical analysis was performed with one-way ANOVA analysis followed by Tukey’s post hoc test. * *p* < 0.05 compared with vehicle control. # *p* < 0.05 compared with docetaxel treatment group.

**Figure 3 ijms-23-13857-f003:**
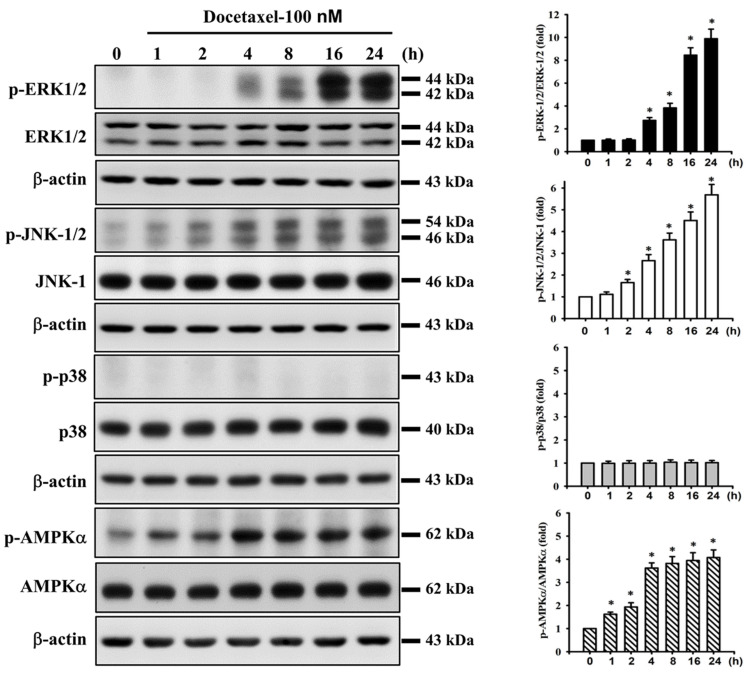
Effect of docetaxel on the protein expression for MAPK and AMPK signals in human tongue squamous carcinoma-derived SAS cells. Cells were treated with docetaxel (100 nM) for 1–24 h. The protein expression of phosphorylated (p)-ERK1/2, ERK, p-JNK1/2, JNK-1, p-p38, p38, p-AMPKα, and AMPKα was determined by Western blot analysis. Results are shown with representative images, and quantification was performed by densitometric analysis. Each presented bar is the mean ± SD of three independent experiments. Statistical analysis was performed with one-way ANOVA analysis followed by Tukey’s post hoc test. * *p* < 0.05 compared with vehicle control.

**Figure 4 ijms-23-13857-f004:**
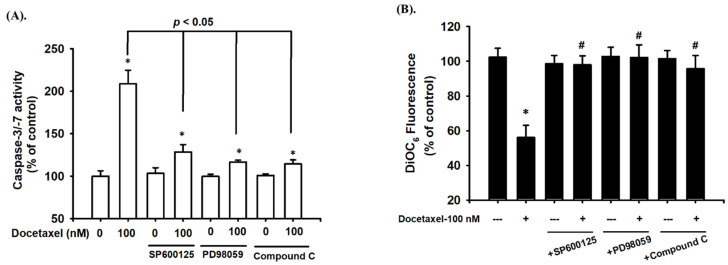
The roles of MAPKs and AMPK in docetaxel-induced apoptotic activity and mitochondrial damage-related signals in human tongue squamous carcinoma-derived SAS cells. Cells were pretreated with SP600125 (a JNK inhibitor; 20 μM), PD98059 (an MEK1/2 inhibitor; 20 μM), and compound c (an AMPK inhibitor; 10 μM) for 1 h and then treated with or without docetaxel for 48 h. (**A**) The caspase-3/-7 activity was determined using a caspase-3/-7 activity assay kit. (**B**) The MMP was determined by flow cytometer with a fluorescent dye DiOC_6_. Data in (**A**,**B**) are presented as the means ± SD of four independent experiments with triplicate determination. Statistical analysis was performed with one-way ANOVA analysis followed by Tukey’s post hoc test. * *p* < 0.05 compared with vehicle control. # *p* < 0.05 compared with docetaxel treatment group.

**Figure 5 ijms-23-13857-f005:**
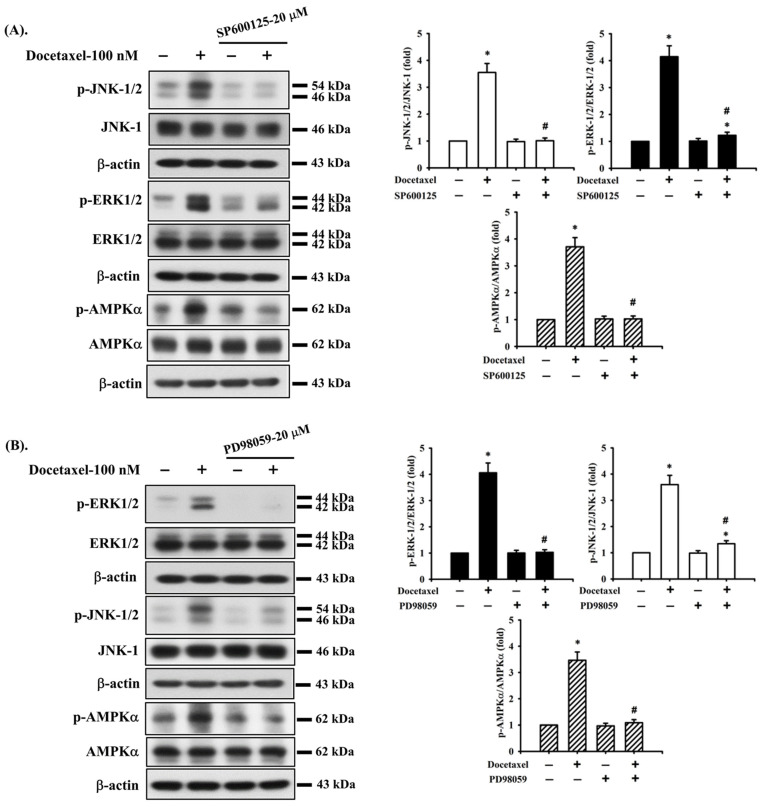
The interaction of ERK, JNK, and AMPK signals in human tongue squamous carcinoma-derived SAS cells. Cells were pretreated with (**A**) SP600125 (a JNK inhibitor; 20 μM), (**B**) PD98059 (an MEK1/2 inhibitor; 20 μM), and (**C**) compound c (an AMPK inhibitor; 10 μM) for 1 h and then treated with or without docetaxel for 8 h. The protein expression of p-JNK1/2, p-ERK1/2, and p-AMPKα was determined by Western blot analysis. Results are shown in (**A**, B, and **C**) of representative images, and quantification was performed by densitometric analysis. Each presented bar is the mean ± SD of three independent experiments. Statistical analysis was performed with one-way ANOVA analysis followed by Tukey’s post hoc test. * *p* < 0.05 compared with vehicle control. # *p* < 0.05 compared with docetaxel treatment group.

**Figure 6 ijms-23-13857-f006:**
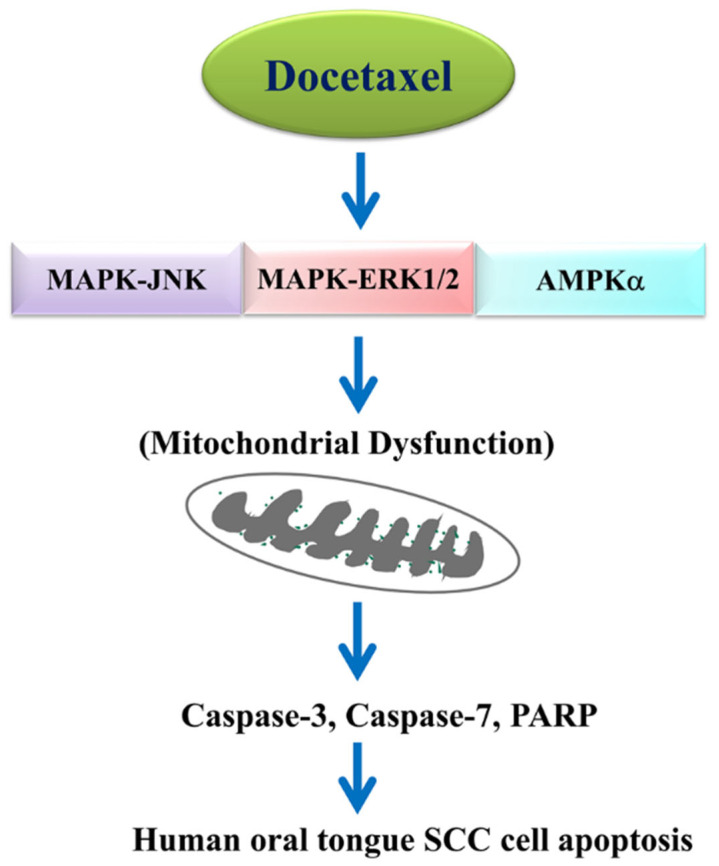
Schematic diagram of the signaling pathways involved in docetaxel-mediated molecular mechanism in oral tongue squamous cell carcinoma cell apoptosis.

## Data Availability

Please contact the corresponding author for reasonable data request.

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
