# Peer review of "Involvement of AMPKα and MAPK-ERK/-JNK Signals in Docetaxel-Induced Human Tongue Squamous Cell Carcinoma Cell Apoptosis"

_ijms, 2022, doi:10.3390/ijms232213857_

Round 1

Reviewer 1 Report

Su et. al demonstrated the novel anti-cancer molecular mechanism of action of docetaxel in tongue squamous cell carcinoma. They found that AMPKa and MAPK-ERK/-JNK signal were involved. This is an interesting finding and give us hints for further investigation for cancer treatment.

1.        In each western blotting results, please show the b-actin results of each protein blotting results. For example, in figure 1-D, the protein expression level of caspase-3, caspase-7 and PARP were showed, but only one b-actin result was showed. There should be three difference b-actin results.

2.        Please labeled the X-axis of the bar graph shown in figure 1D, 3

3.        In figure 1D, 2B, why the treatment period is difference between the blotting results and the quantification results? Please correct.

4.        In figure 1D, 2B, please the describe the meaning of **

5.        In figure 2B, why the treatment period is difference between the blotting results and the quantification results? Please correct.

6.        Which antibiotics were used during cell culture? Please describe in your manuscript.

7.        Did the docetaxel-mediated molecular mechanism can be observed in other tongue cancer cell lines? Please exclude the cell line specific issue.

8.        Please correct the title of the 2.2

9.        In Figure 2C and 2D, authors showed docetaxel upregulated the mRNA level of Bax, Bak, Bad,and p53. Whether docetaxel affect the protein expression level of Bax, Bak, Bad, p53, and BCL2?

10.     Authors showed that the docetaxel-induced caspase3/7 activity was inhibited when co-treating with JNK, ERK, and AMPK inhibitors. Whether this inhibitory effect could be observed after JNK, ERK, and AMPK-specific targeting treatment such as siRNA transfection.

Author Response

Reply to Reviewer’ 1 comments:

Q1. In each western blotting results, please show the b-actin results of each protein blotting results. For example, in figure 1-D, the protein expression level of caspase-3, caspase-7 and PARP were showed, but only one b-actin result was showed. There should be three difference b-actin results.

 Response: We appreciate the reviewer's comments.

According to reviewer’s suggestion, we have added the b-actin results of ach protein blotting results in the ‘‘2. Results’’ section of the revised manuscript.

Q2. Please labeled the X-axis of the bar graph shown in figure 1D, 3

Response: The label of X-axis of the bar graph in Figure 1D have added.

Q3. In figure 1D, 2B, why the treatment period is difference between the blotting results and the quantification results? Please correct.

Response: These mistakes have corrected.

Q4. In figure 1D, 2B, please the describe the meaning of **

Response:

  1. The mark ‘‘**’’, as the same with ‘‘*’’, is mean p value < 0.05.
  2. These misunderstanding markers in Figure 1B, 2A, and 2D have been corrected to ‘‘*’’.  

Q5. In figure 2B, why the treatment period is difference between the blotting results and the quantification results? Please correct.

Response: This mistake has corrected.

Q6. Which antibiotics were used during cell culture? Please describe in your manuscript.

Response: The information of antibiotics used during cell culture have added in the manuscript. These descriptions in the ‘4.2. Cell culture’ were shown as follows:

‘‘Cells were maintained in culture medium containing 45% Dulbecco’s modified Eagle’s medium, 45% Ham’s F12 medium supplemented with 10% fetal bovine serum (FBS), 1% penicillin-streptomycin, and 0.5% gentamicin (all from Gibco; Thermo Fisher Scientific, Inc.).’’ in L343-L346 of the revised manuscript.

Q7. Did the docetaxel-mediated molecular mechanism can be observed in other tongue cancer cell lines? Please exclude the cell line specific issue.

Response: In our study, we used human tongue cancer SAS cells in experiments. SAS cells is the most common cells used in oral tongue SCC investigation. Many studies also used SAS cells investigated the possible molecular mechanisms and therapeutic agents in oral tongue SCC. For example, a study showed capsaicin, which is a component of chill peppers, induced autophagy through tumor associated NADH oxidase (Chang et al., 2020). Another study reported quercetin, a polyphenolic flavonoid, exert a possible anticancer agent through induced apoptosis (Ma et al., 2018). Moreover, a study used gene transfected SAS cells to construct a xenograft mouse model in oral tongue SCC investigation (Chen et al., 2021). However, there are other tongue cancer SAS cells could be used in oral cancer study, such as SCC-4 cells (CRL-1624, ATCC), CAL-27 (CRL2095, ATCC), HSC-3 (SCC-193, Merck-Sigma-Aldrich), YD-8 (ACH-000630, depmap portal). These cells description above will provide us more cells model to demonstrate detailed molecular mechanism in tongue SCC therapy in the future works.

We described a limitation for this issue in the ‘‘3.Discussion’’section of this revised manuscript.

References:

  1. Chang, C.F., Islam, A., Liu, P.F., Zhan, J.H., Chueh, P.J. Capsaicin acts through tNOX (ENOX2) to induce autophagic apoptosis in p53-mutated HSC-3 cells but autophagy in p53-functional SAS oral cancer cells. Am J Cancer Res 2020, 10, 3230-3247.
  2. Chen, Y.A., Ho, C.L., Ku, M.T., Hwu, L., Lu, C.H., Chiu, S.J., Chang, W.Y., Liu, R.S. Detection of cancer stem cells by EMT-specific biomarker-based peptide ligands. Sci Rep 2021, 11,
  3. Ma, Y.S., Yao, C.N., Liu, H.C., Yu, F.S., Lin, J.J., Lu, K.W., Liao, C.L., Chueh, F.S., Chung, J.G. Quercetin induced apoptosis of human oral cancer SAS cells through mitochondria and endoplasmic reticulum mediated signaling pathways. Oncol Lett 2018, 15, 9663-9672.

Q8. Please correct the title of the 2.2

Response: The title of the 2.2. has corrected to ‘‘2.2. Docetaxel-induced apoptosis is mediated by a mitochondria-dependent pathway in human tongue SCC SAS cells’’.

Q9. In Figure 2C and 2D, authors showed docetaxel upregulated the mRNA level of Bax, Bak, Bad,and p53. Whether docetaxel affect the protein expression level of Bax, Bak, Bad, p53, and BCL2?

Response:

  1. The protein expression levels of Bcl-2 and Bax were shown in Figure 4C-4E.
  2. We have added the sentence to explain ‘‘the effect of protein expression levels of Bcl-2 and Bax in docetaxel-treated SAS cells’’. These descriptions in the ‘2. Docetaxel-induced apoptosis is mediated by a mitochondria-dependent pathway in human tongue SCC SAS cells’ section were shown as follows:

‘‘Furthermore, docetaxel significantly increased pro-apoptotic Bax, Bak, Bad, and p53 (Figure 2C) and decreased anti-apoptotic Bcl-2 (Figure 2D) gene expression levels. As shown in Figure 4C-4E, treatment with docetaxel for 48 h also significantly de-creased Bcl-2 and increased Bax protein expression in SAS cells. These showed a significant shift in the pro-apoptotic/anti-apoptotic ratio toward a state associated with apoptosis. These results indicate that mitochondria-dependent apoptotic pathway plays an important role in docetaxel-induced SAS cell death.’’ (in L141-L147 of the revised manuscript).

Q10. Authors showed that the docetaxel-induced caspase3/7 activity was inhibited when co-treating with JNK, ERK, and AMPK inhibitors. Whether this inhibitory effect could be observed after JNK, ERK, and AMPK-specific targeting treatment such as siRNA transfection.

Response:

    In this study, docetaxel induced tongue SCC apoptosis through an interdependent MAPK-JNK, MAPK-ERK1/2, and AMPKα signaling pathway. We used specific inhibitors, including SP600125 (selective JNK inhibitor), PD98059 (selective ERK inhibitor), and compound c (selective AMPK inhibitor), to clarify the relationships of MAPK-JNK, MAPK-ERK1/2, and AMPKα affected apoptosis in docetaxel treated SAS cells. The SP600125, PD98059 and compound c is widely used in many studies, which discussed about MAPK and AMPK pathways (Chuang et al., 2014; Liu et al., 2014; Fong et al., 2017). In human non-small cell lung cancer study, SP600125 and PD98059 was used to validate the MAPK involved in quinoline analog induced apoptosis (Fong et al., 2017). Moreover, compound c, which is a cell-permeable AMPK inhibitor, is used to clarify the AMPK regulated anticancer signaling (Liu et al., 2014; Chiang et al., 2017). Especially in oral tongue SCC, SP600125 and PD98059 was used to demonstrate the MAPK involved in ROS activated caspases signals in anti-cancer agent (Jo et al., 2012; Kim et al., 2013). Moreover, SP600125 was used to demonstrate the IL-6 regulated oral SCC migration (Chuang et al., 2014). Moreover, compound c has been used to demonstrate the AMPK involved in ROS induced autophagy in oral SCC (Xu et al., 2017).

    In references, there are other experimental methods to help us clarify the relationships of MAPK-JNK, MAPK-ERK1/2, and AMPKα in docetaxel induced SAS cells apoptosis. For example, a study used short hairpin RNA (shRNA)-JNK to knockdown of JNK to demonstrate the JNK regulated cantharidin induced OSCC apoptosis (Su et al., 2016). Moreover, the small interfering RNA (siRNA) silencing of JNK1/2 was used to give a suggestion that inhibit JNK1/2 might be a therapeutic implication of oral SCC (Gkouveris et al., 2016). Thus, we can use shRNA and siRNA to clarify more distinct molecular mechanisms in oral tongue SCC investigation.

   We described a limitation for this issue in the Discussion section of this revised manuscript.

References:

  1. Chuang, J.Y., Huang, Y.L., Yen, W.L., Chiang, I.P., Tsai, M.H., Tang, C.H. Syk/JNK/AP-1 signaling pathway mediates interleukin-6-promoted cell migration in oral squamous cell carcinoma. Int J Mol Sci 2014, 15, 545-559.
  2. Liu, X., Chhipa, R.R., Nakano, I., Dasgupta, B. The AMPK inhibitor compound C is a potent AMPK-independent antiglioma agent. Mol Cancer Ther 2014, 13, 596-605.
  3. Fong, Y., Wu, C.Y., Chang, K.F., Chen, B.H., Chou, W.J., Tseng, C.H., Chen, Y.C., Wang, H.D., Chen, Y.L., Chiu, C.C. Dual roles of extracellular signal-regulated kinase (ERK) in quinoline compound BPIQ-induced apoptosis and anti-migration of human non-small cell lung cancer cells. Cancer Cell Int 2017, 17, 37.
  4. Chiang, C.F., Chao, T.T., Su, Y.F., Hsu, C.C., Chien, C.Y., Chiu, K.C., Shiah, S.G., Lee, C.H., Liu, S.Y., Shieh, Y.S. Metformin-treated cancer cells modulate macrophage polarization through AMPK-NF-kappaB signaling. Oncotarget 2017, 8, 20706-20718.
  5. Jo, J.R., Park, J.S., Park, Y.K., Chae, Y.Z., Lee, G.H., Park, G.Y., Jang, B.C. Pinus densiflora leaf essential oil induces apoptosis via ROS generation and activation of caspases in YD-8 human oral cancer cells. Int J Oncol 2012, 40, 1238-1245.
  6. Kim, J.Y., An, J.M., Chung, W.Y., Park, K.K., Hwang, J.K., Kim du, S., Seo, S.R., Seo, J.T. Xanthorrhizol induces apoptosis through ROS-mediated MAPK activation in human oral squamous cell carcinoma cells and inhibits DMBA-induced oral carcinogenesis in hamsters. Phytother Res 2013, 27, 493-498.
  7. Xu, Z., Huang, C.M., Shao, Z., Zhao, X.P., Wang, M., Yan, T.L., Zhou, X.C., Jiang, E.H., Liu, K., Shang, Z.J. Autophagy Induced by Areca Nut Extract Contributes to Decreasing Cisplatin Toxicity in Oral Squamous Cell Carcinoma Cells: Roles of Reactive Oxygen Species/AMPK Signaling. Int J Mol Sci 2017, 18(3), 524.
  8. Su, C.C., Lee, K.I., Chen, M.K., Kuo, C.Y., Tang, C.H., Liu, S.H. Cantharidin Induced Oral Squamous Cell Carcinoma Cell Apoptosis via the JNK-Regulated Mitochondria and Endoplasmic Reticulum Stress-Related Signaling Pathways. PLoS One 2016, 11, e0168095.
  9. Gkouveris, I., Nikitakis, N., Karanikou, M., Rassidakis, G., Sklavounou, A. JNK1/2 expression and modulation of STAT3 signaling in oral cancer. Oncol Lett 2016, 12, 699-706.

Reviewer 2 Report

In this study, the authors investigate the effect of docetaxel on cell apoptosis in a human tongue SCC-derived cell line and show that docetaxel can induce a series of apoptosis-related events in SAS cells via an interdependent MAPK-JNK, MAPK-ERK1/2, and AMPK signaling pathways, including cytotoxicity, mitochondrial damage and phosphorylation of ERK, JNK, and AMPK signaling molecules. The study is interesting, however, there are a few concerns about the quality of the manuscript in its current form.

The major limitation of the study is that the authors only analysed one cell line. Usually, at least two cell lines should be assessed in the study.

Statistical analysis: Throughout the figures, the authors showed the same p values (i.e. * or **) between different group comparisons in the same graph, although big differences were shown between them. Were they all having the same range of the p values based on ANOVA followed by Tukey’s post hoc test as described in M+M?

The description of what stat methods were used to obtain the p values should be stated in each figure legend independently.

Fig.2 C&D: why no Western blotting data are shown here? The protein expression is crucial to demonstrate cellular apoptosis.    

The MS needs further proofreading as there are sentences which are not well written, e.g. 1) in all legends “All results shown on representative images, …”; 2) 2.2. Norketamine (NK)-induced apoptosis is mediated by a mitochondria-dependent pathway in RT4 cells. What is this? 3) ‘Yew’ or ‘yew’ and ‘and’ in the authors, etc.

Ideally, the authors should summarise the major findings in a schematic diagram. 

The original data should be the whole membranes rather than the same ones shown in the figures.

Author Response

Reply to Reviewer’ 2 comments:

Q1. The major limitation of the study is that the authors only analysed one cell line. Usually, at least two cell lines should be assessed in the study.

Response: In our study, we used human tongue cancer SAS cells in experiments. SAS cells is the most common cells used in oral tongue SCC investigation. Many studies also used SAS cells investigated the possible molecular mechanisms and therapeutic agents in oral tongue SCC. For example, a study showed capsaicin, which is a component of chill peppers, induced autophagy through tumor associated NADH oxidase (Chang et al., 2020). Another study reported quercetin, a polyphenolic flavonoid, exert a possible anticancer agent through induced apoptosis (Ma et al., 2018). Moreover, a study used gene transfected SAS cells to construct a xenograft mouse model in oral tongue SCC investigation (Chen et al., 2021). However, there are other tongue cancer SAS cells could be used in oral cancer study, such as SCC-4 cells (CRL-1624, ATCC), CAL-27 (CRL2095, ATCC), HSC-3 (SCC-193, Merck-Sigma-Aldrich), YD-8 (ACH-000630, depmap portal). These cells description above will provide us more cells model to demonstrate detailed molecular mechanism in tongue SCC therapy in the future works.

We described a limitation for this issue in the ‘‘3.Discussion’’section of this revised manuscript.

References:

  1. Chang, C.F., Islam, A., Liu, P.F., Zhan, J.H., Chueh, P.J. Capsaicin acts through tNOX (ENOX2) to induce autophagic apoptosis in p53-mutated HSC-3 cells but autophagy in p53-functional SAS oral cancer cells. Am J Cancer Res 2020, 10, 3230-3247.
  2. Chen, Y.A., Ho, C.L., Ku, M.T., Hwu, L., Lu, C.H., Chiu, S.J., Chang, W.Y., Liu, R.S. Detection of cancer stem cells by EMT-specific biomarker-based peptide ligands. Sci Rep 2021, 11,
  3. Ma, Y.S., Yao, C.N., Liu, H.C., Yu, F.S., Lin, J.J., Lu, K.W., Liao, C.L., Chueh, F.S., Chung, J.G. Quercetin induced apoptosis of human oral cancer SAS cells through mitochondria and endoplasmic reticulum mediated signaling pathways. Oncol Lett 2018, 15, 9663-9672.

Q2. Statistical analysis: Throughout the figures, the authors showed the same p values (i.e. * or **) between different group comparisons in the same graph, although big differences were shown between them. Were they all having the same range of the p values based on ANOVA followed by Tukey’s post hoc test as described in M+M? The description of what stat methods were used to obtain the p values should be stated in each figure legend independently.

Response:  

  1. The mark ‘‘**’’, as the same with ‘‘*’’, is mean p value < 0.05.
  2. These misunderstanding markers in Figure 1B, 2A, and 2D have been corrected to ‘‘*’’.
  3. We have added the description about the stat method used to obtain the p values in figure legend.

Q3. Fig.2 C&D: why no Western blotting data are shown here? The protein expression is crucial to demonstrate cellular apoptosis.    

Response:

  1. The protein expression levels of Bcl-2 and Bax were shown in Figure 4C-4E.
  2. We have added the sentence to explain ‘‘the effect of protein expression levels of Bcl-2 and Bax in docetaxel-treated SAS cells’’. These descriptions in the ‘2. Docetaxel-induced apoptosis is mediated by a mitochondria-dependent pathway in human tongue SCC SAS cells’ section were shown as follows:

‘‘Furthermore, docetaxel significantly increased pro-apoptotic Bax, Bak, Bad, and p53 (Figure 2C) and decreased anti-apoptotic Bcl-2 (Figure 2D) gene expression levels. As shown in Figure 4C-4E, treatment with docetaxel for 48 h also significantly de-creased Bcl-2 and increased Bax protein expression in SAS cells. These showed a significant shift in the pro-apoptotic/anti-apoptotic ratio toward a state associated with apoptosis. These results indicate that mitochondria-dependent apoptotic pathway plays an important role in docetaxel-induced SAS cell death.’’ (in L141-L147 of the revised manuscript).

Q4. The MS needs further proofreading as there are sentences which are not well written, e.g. 1) in all legends “All results shown on representative images, …”; 2) 2.2. Norketamine (NK)-induced apoptosis is mediated by a mitochondria-dependent pathway in RT4 cells. What is this? 3) ‘Yew’ or ‘yew’ and ‘and’ in the authors, etc.

Response: We appreciate the reviewer's comments.

We have been carefully corrected and proofreading our revised manuscript. Moreover, the language usage, grammar and syntax of our manuscript has also been revised by an English Editing Services of MDPI Author Services (English Editing ID english-53753, and as below shown ‘the certification of English-editing of revised manuscript’).

Q5. Ideally, the authors should summarise the major findings in a schematic diagram. 

Response: We summarized the major findings in a schematic diagram (Figure 6) in this revised manuscript according to the suggestion of reviewer.

Q6. The original data should be the whole membranes rather than the same ones shown in the figures.

Response: The original data for Western blot presented have been provided.

Reviewer 3 Report

The authors aimed to evaluate the roles of AMPKα and MAPKs signals in human oral tongue SCC cell growth, and to determine whether docetaxel could reduce oral tongue SCC cell growth through regulating the AMPKα and MAPKs signals.

The study covers some issues that have been overlooked in other similar topics. The structure of the manuscript appears adequate and well divided in the sections. Moreover, the study is easy to follow, but some issues should be improved. Some of the comments that would improve the overall quality of the study are:

a. Authors must pay attention to the technical terms acronyms they used in the text.

b. Limitations of the study needs to be added.

c. Conclusion Section: This paragraph required a general revision to eliminate redundant sentences and to add some "take-home message".

Author Response

Reply to Reviewer’ 3 comments:

Q1. Authors must pay attention to the technical terms acronyms they used in the text.

Response: We appreciate the reviewer's comments.

We have been carefully corrected and proofreading our revised manuscript (including the technical terms acronyms). Moreover, the language usage, grammar and syntax of our manuscript has also been revised by an English Editing Services of MDPI Author Services (English Editing ID english-52753, and as below shown ‘the certification of English-editing of revised manuscript’).

Q2. Limitations of the study needs to be added.

Response: We have added the limitations of the study in the end of the Discussion section of this revised manuscript according to the suggestion of reviewer.

Limitations of the study: (1) A human tongue cancer SAS cell line was used in this study. Other tongue cancer cell lines can further confirm the docetaxel-mediated molecular mechanisms. (2) The pharmacological inhibitors for JNK, ERK, and AMPK were used in this study. The JNK, ERK, and AMPK-specific targeting gene silence can further confirm the docetaxel-mediated molecular mechanisms. (3) The future animal model studies can further support the findings from the in vitro cancer cell model. (in L312-L318 of the revised manuscript).

Q3. Conclusion Section: This paragraph required a general revision to eliminate redundant sentences and to add some "take-home message".

Response: We appreciate the reviewer's comments.

The paragraph of ‘‘Conclusion’’ has reorganized and revised, as follows:

  1. Conclusions

The results of this study, as shown in Figure 6, demonstrate that interdepend-ent MAPK-JNK, MAPK-ERK1/2, and AMPKα signaling pathways are involved in docetaxel-induced mitochondria-dependent human tongue SCC cell apoptosis and death. These in vitro findings suggest that docetaxel possesses pharmacological potential for oral tongue SCC cancer therapy. The findings of cell model studies may be further explored in future animal model studies to elucidate the mechanism of action of docetaxel. (in L468-L474 of the revised manuscript).

Round 2

Reviewer 2 Report

Please correct the following sentences:

Pg 2 line 81: AMPK can be activated by AMP binding or phosphorylation of the α subunit at residue Thr-172 residue phosphorylation [25, 26].

Pg 9 line 243: There is a disturbing trend of increasing tongue SCC is increased in younger patients, including …

Pg 14 Line 480: Figure 6. T Schematic diagram of the signaling pathways involved in docetaxel-mediated …

In addition, in Figures 1D, 4C-E & 5A-C, the text labels on the X-axis are too small to be legible. Please make them larger.

Author Response

Reply to Reviewer’ 2 comments:

Q1. Pg 2 line 81: AMPK can be activated by AMP binding or phosphorylation of the α subunit at residue Thr-172 residue phosphorylation [25, 26].

Response: This sentence has been corrected and rewritten to ‘‘It has been reported that the decrease of cellular ATP would cause the AMP/ATP ratio to increase. This effect resulted to the AMP binding to AMPK which caused the upstream kinases, such as LKB1 and CAMKKβ, to phosphorylate the AMPKα subunits on the Thr172 [25-28].’’ (in L81-L84 of the revised manuscript).

Q2. Pg 9 line 243: There is a disturbing trend of increasing tongue SCC is increased in younger patients, including …

Response: This sentence has been corrected and rewritten to ‘‘A report showed the incidence of tongue SCC increasing at the age of 18 to 44 of young white women [36].’’ (in L250-L251 of the revised manuscript).

Q3. Pg 14 Line 480: Figure 6. T Schematic diagram of the signaling pathways involved in docetaxel-mediated …

Response: This sentence has been corrected to ‘‘Figure 6. Schematic diagram of the signaling pathways involved in docetaxel-mediated molecular mechanism in oral tongue squamous cell carcinoma cell apoptosis.’’ (in L483-L484 of the revised manuscript).

Q4. In addition, in Figures 1D, 4C-E & 5A-C, the text labels on the X-axis are too small to be legible. Please make them larger.

Response: We have according to reviewer’s suggestion to enlarge the labels on the X-axis of Figures 1D, 2B, 4C-E, and 5A-C.
